# A study protocol for the validation of a prognostic model with an emphasis on modifiable factors to predict chronic pain after a new episode of acute- or subacute nonspecific idiopathic, non-traumatic neck pain presenting in primary care

**Martine J. Verwoerd**[1]*, **Harriet Wittink**[1], **Francois Maissan**[1], **Sander M. J. van Kuijk**[2], **Rob J. E. M. Smeets**[3]

1 Research Group Lifestyle and Health, Utrecht University of Applied Sciences, Utrecht, The Netherlands, 2 Department of Clinical Epidemiology and Medical Technology Assessment, Maastricht University Medical Centre, Maastricht, The Netherlands, 3 Department of Rehabilitation Medicine, Research School CAPHRI, Maastricht University, CIR Rehabilitation, Eindhoven, The Netherlands, Pain in Motion International Research Group (PiM), www.paininmotion.be

* martine.verwoerd@hu.nl

## Abstract

### Background

The primary objective of this study is to identify which modifiable and non-modifiable factors are independent predictors of the development of chronic pain in patients with acute- or subacute nonspecific idiopathic, non-traumatic neck pain, and secondly, to combine these to develop and internally validate a prognostic prediction model.

### Methods

A prospective cohort study will be conducted by physiotherapists in 30 primary physiotherapy practices between January 26, 2020, and August 31, 2022, with a 6-month follow-up until March 17, 2023. Patients who consult a physiotherapist with a new episode of acute- (0 to 3 weeks) or subacute neck pain (4 to 12 weeks) will complete a baseline questionnaire. After their first appointment, candidate prognostic variables will be collected from participants regarding their neck pain symptoms, prior conditions, work-related factors, general factors, psychological and behavioral factors. Follow-up assessments will be conducted at six weeks, three months, and six months after the initial assessment. The primary outcome measure is the Numeric Pain Rating Scale (NPRS) to examine the presence of chronic pain. If the pain is present at six weeks, three months, and six months with a score of NPRS $\geq 3$, it is classified as chronic pain. An initial exploratory analysis will use univariate logistic regression to assess the relationship between candidate prognostic factors at baseline and outcome. Multiple logistic regression analyses will be conducted. The discriminative ability of the prognostic model will be determined based on the Area Under the receiver operating

**Data Availability Statement:** Data sharing is not applicable to this article as the current study describes the protocol for the data analysis. For the results paper, the datasets generated during and/or analysed during this study will be made available in an open access repository.

**Funding:** This PhD trajectory and research is partly supported by the Institute of Movement Studies and partly by the Utrecht University of Applied Sciences research voucher. The funding concerns am internal promotion voucher of the University of Applied Sciences. The funders had no role in the study design, data collection, analysis, decision to publish, or manuscript preparation.

**Competing interests:** The authors have declared that no competing interests exist.

characteristic Curve (AUC), calibration will be assessed using a calibration plot and formally tested using the Hosmer and Lemeshow goodness-of-fit test, and model fit will be quantified as Nagelkerke's $R^2$. Internal validation will be performed using bootstrapping-resampling to yield a measure of overfitting and the optimism-corrected AUC.

## Discussion

The results of this study will improve the understanding of prognostic and potential protective factors, which will help clinicians guide their clinical decision making, develop an individualized treatment approach, and predict chronic neck pain more accurately.

## Introduction

Neck pain is one of the most prevalent and disabling health conditions, with a substantial impact on public health [1, 2]. The Global Burden of Disease study demonstrated that neck pain is third in the ranking of 'years lived with disability' in non-fatal diseases in Europe [3]. Costs related to neck pain are rising mainly due to extended work absence and usage of health care services [1, 4, 5]. In particular, neck pain that becomes chronic causes high healthcare costs [6]. The prevalence of chronic neck pain has increased from 2005 to 2015 by 21% up to approximately 358 million people worldwide, and it is likely to increase further in Western countries due to an aging population [7]. In the Netherlands, pain in the cervical region is the most commonly reported complaint for which patients seek help in physiotherapy practices [8].

Recovery from neck pain and related disability mainly occurs in the first few weeks. Thereafter, the recovery rate is much lower [9, 10]. The reported effect of physiotherapy treatment in patients with chronic musculoskeletal pain is, at best, only moderate [11–13]. It is therefore not surprising that defining the natural course and the prognostic factors in people with acute- and subacute neck pain is a top-five priority of the new agenda for Neck Pain Research [14]. Knowledge of prognostic factors can help health care providers to improve clinical decision-making and is a likely key factor in combatting chronification of idiopathic neck pain. Preventing chronicity should be the major focus of physiotherapists in the (sub)acute phase of musculoskeletal pain. Being able to predict which patients with neck pain are likely to develop chronic pain may help prevent chronification of pain in physiotherapy practices.

At the present time the existing literature on prognostic models shows a low performance in predicting chronicity or recovery from neck pain [15, 16], it is thereby not applicable as a starting point for a new prognostic study. A limitation and possible explanation of this low performance is the inclusion of a too-heterogeneous group of neck pain patients. Most studies include (sub)acute neck pain, whiplash-related neck pain, pain with neurological symptoms, and even patients who already have chronic pain [15, 17, 18], although these groups are known to differ in both clinical symptoms and prognosis [19–21]. Therefore, it seems useful to pay attention to the pain etiology and pathophysiological mechanisms of the existent pain in classification and inclusion systems [22].

In addition, prognostic research has often focused on factors that are non-modifiable by physiotherapists, such as age and sex [16]. Only clinically modifiable factors have the potential to change patient outcome and are therefore recommended to be included in prognostic research [16, 23]. However, to strengthen a prognostic model, it can be relevant to include some non-modifiable factors. Based on a recent consensus study of potential modifiable

prognostic factors, including psychosocial factors in prognostic research for chronification is relevant [24]. It seems that psychosocial factors in particular can be modified. Furthermore, it is known that neurophysiological changes in the chronification of pain are modulated by psychosocial factors [25].

Therefore, there is a need for a prognostic study that identifies modifiable prognostic factors using a biopsychosocial view, that includes only patients with acute- (0 to 3 weeks) or subacute (4 to 12 weeks) nonspecific idiopathic, non-traumatic neck pain, to help prevent chronification of pain in physiotherapy practices. This study should occur in primary care physiotherapy practices and with a cohort of patients of an adequate sample size.

The primary objective of this study is to identify which modifiable and non-modifiable factors are independent predictors of the development of chronic pain in patients with acute- or subacute nonspecific idiopathic, non-traumatic neck pain, and secondly, to combine these to develop and internally validate a prognostic prediction model.

## Methods

### Study design

The present study is a prospective cohort study of prognostic factors informed by the PROG-RESS framework and TRIPOD statement type 1b and specific recommendations for statistical approaches to Type 3 prognostic model research [26, 27]. This study will be reported in accordance with the Transparent Reporting of a multivariable prediction model for Individual Prognosis Or Diagnosis (TRIPOD) statement [27].

### Study setting

Potential participants will be selected from 30 primary care physiotherapy practices including 81 physiotherapists between January 26, 2020, and August 31, 2022, and is due to be completed at March 17, 2023 (including reminders and time for response).

For the generalizability of this research, we selected physiotherapists with different backgrounds; physiotherapists pursuing a master's degree working in primary care and experienced physiotherapists with and without affiliation to an academic institute will include participants.

### Ethical approval

The Medical Research Ethics Committee approved that this study (protocol number: 19-766/C) does not apply to the Medical Research Involving Human Subjects Act (WMO). Therefore an official approval of this study by the Medical-Ethical Review Committee (METC) Utrecht is not required under the WMO Utrecht. All data is processed anonymously, and all participants have to sign an informed consent. The participants receive a personal code upon inclusion, which must be submitted at each measurement moment. The measurements will be collected through the secure data transfer system Formdesk [28].

### Participants

The patients will be approached if they present with a new episode of acute- (0 to 3 weeks) or subacute (4 to 12 weeks) nonspecific idiopathic, non-traumatic neck pain. To be eligible to take part in the study, participants must meet the following criteria:

1. The patients are at least 18 years or older.

2. The patients have a new presentation of neck pain not more than 12 weeks upon onset.

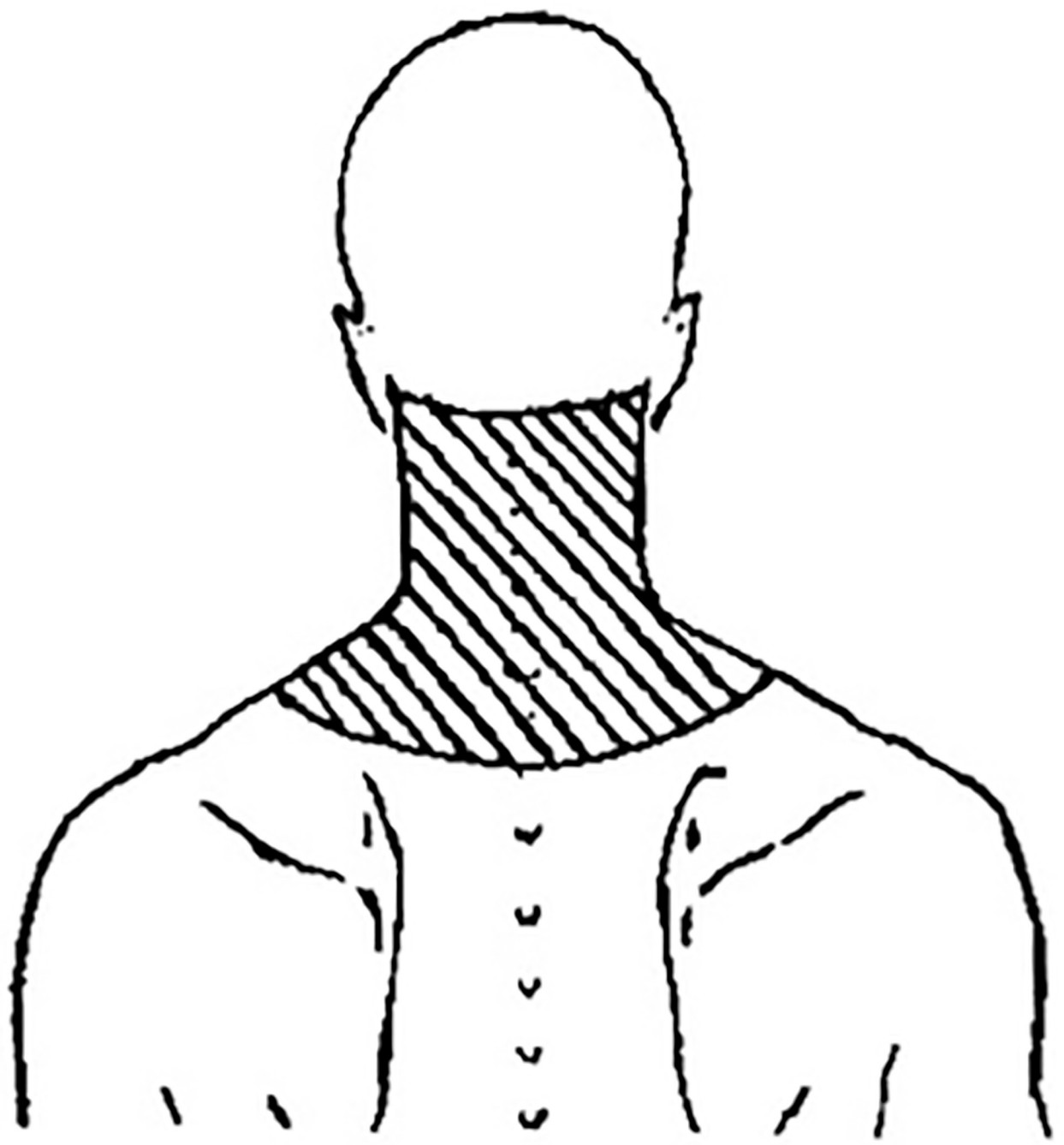

**Fig 1. Neck pain area used for inclusion [31].**

3. The neck pain region has to fall within the used region presented in Fig 1.

4. If the patient has had neck pain before, the patients must be relatively free from symptoms for at least three months (Numeric Pain Rating Scale (NPRS) of < 3) prior to this new episode of neck pain.

These inclusion criteria will effectively exclude the population with chronic pain [29, 30].

The following general and specific exclusion criteria will be examined at an initial history taking by the physiotherapist prior to the recruitment:

**Specific exclusion criteria.**

1. Neck surgery in the past.

2. Cervical spine radiculopathy measured with the Upper Limb Neurodynamic Test 1 [32].

3. Widespread pain (ICD 11); diffuse musculoskeletal pain in at least 4 of 5 body regions and in at least 3 or more body quadrant (as defined by upper-lower / left-right side of the body) and axial skeleton (neck, back, chest, and abdomen).

4. Pain not caused by a musculoskeletal origin (not located in in the muscles, bones, joints, or tendons) [33].

**General exclusion criteria.**

1. Inability to read or understand the Dutch language.

The participating physiotherapists record reasons for exclusion during the study period. In addition, an anonymized record will be kept of patients who meet the inclusion criteria but choose not to participate and their reasons for doing so. The treatment the patients receive will be reported. The coding will be done based on the Dutch Physiotherapy Guideline for neck pain [34, 35]. Participation in this study has no influence on the content of the treatment.

## Baseline and follow-up procedure

If the patient meets the criteria during the first consultation, the physiotherapist informs the patient orally about the purpose and discusses participant expectations of the study. If the patient indicates verbally that he/she wants to participate in the study, written informed consent is obtained from the participant before the first questionnaire is completed. Subsequently, each participant receives a digital questionnaire sent via a link by email in week one (T0, baseline) and at six weeks (T1), three months (T2), and six months (T3). The T0 questionnaire takes 30–40 minutes to complete, the T1 measurement 20–30 minutes, and the T2 and T3 around 20 minutes. If the participant has not completed a questionnaire after one week, a reminder is sent by email or telephone contact will be made by the therapist who includes the participant. This procedure is repeated one week later, if necessary.

## Outcome

The NPRS is used to quantify the presence of chronic pain. If pain is present at all measurement moments, six weeks, three months, and six months with a score of NPRS $\geq$3, it will be classified as chronic pain [30, 36]. The NPRS is known to have an average reliability (ICC = 0.67 [0.27–0.84]) in neck pain, the minimal detectable change is 2.6 and a minimum clinically important difference of 1.5 in patients with mechanical neck pain [37]. The NPRS is an inventory and evaluation questionnaire, which was found to be valid [38].

## Candidate prognostic factors

The candidate prognostic factors are based on our previous systematic review and Delphi study [16, 24]. From the systematic review, we included the variables significantly predictive of

pain chronification or non-recovery. Furthermore, we included the variable with a consensus of >70% in the first round of our Delphi study.

Table 1 shows the researched domains, candidate prognostic factors, the measure method used and how the variables will be handled in the statistical analysis.

**Symptoms.** The symptoms are current pain intensity (measured with the NPRS), duration of the neck pain in weeks, and whether the patient experiences pain in multiple body regions, all measured with a single question. Duration of pain will be handled as a continuous variable in our statistical analysis since there is no hard cut-off point between 'acute' and 'subacute' pain. Headaches are surveyed using a three-categorical single question to dichotomize specifically 'headaches that originated together with neck pain' and 'no headaches or headaches that exist before the neck pain'.

The Pain Disability Index (PDI) is a 7-item questionnaire that investigates the extent of self-reported pain-related disability [39]. The PDI measures family/home responsibilities, recreation, social activity, occupation, sexual behavior, self-care, and life support. The questionnaire items are assessed on a 0–10 numeric rating scale in which 0 means no disability and 10 is maximum disability.

**Work related factors.** The questions about happiness at work, job satisfaction, and the potential to self-modify posture during work are non-validated questions of which the psychometric properties are unknown and have been developed and formulated based on a Delphi study [24]. These are all answered on a Likert scale (1–5), which will be dichotomized in the statistical analysis (Table 1).

**General factors.** Lifestyle is measured with self-reported questions on different lifestyle domains; physical activity, smoking, alcohol, weight, and sleep quality.

Sleep quality is questioned through an adjusted question from the Neck Disability Index (NDI). The question was adjusted based on a Delphi study, which indicates that the NDI does not sufficiently question the 'sleep quality' factor [24]. For this reason, the statements "I do not wake up in the morning rested" and "I have trouble falling asleep" were added to the existing 9[th] question of the NDI questionnaire [47]. Since the question was modified, no psychometric properties are known.

**Psychological and behavior factors.** Catastrophizing is measured with a shortened validated 6-item version of the Pain Catastrophizing Scale (PCS) that assesses catastrophic thoughts or feelings associated with the experience of pain. Participants are asked to think about a recent painful experience and indicate to what extent they experience each of the six thoughts or feelings when they are in pain. The short version of the PCS assesses each dimension to capture the broad construct of catastrophizing; it compromises the lower-order factors labeled as rumination, magnification, and helplessness [40]. It uses a 5-point scale ranging from 0 (not at all) to 4 (always) [48]. A shortened version of the PCS is used to limit the total measurement duration. Internal, construct, and the smallest detectable change (SDC) are highly comparable to the original PCS [40].

Kinesiophobia is measured using the Tampa Scale for Kinesiophobia 11-item version (TSK-11). This short version assesses both dimensions of kinesiophobia; harm and activity avoidance. The eleven questions are scored from 1 (strongly disagree) to 4 (strongly agree). The psychometric properties of the TSK-11 demonstrate good internal consistency ($\alpha = 0.79$), responsiveness (SRM = -1,11), test-retest reliability (ICC = 0,81, SEM = 2.54), concurrent validity and predictive validity [43].

In a (sub)acute state of pain, a response such as fear of movement or negative orientation toward pain could exist. However, it is not known when this response is a beneficial level of adaptation or an excessive response to (sub)acute pain. Furthermore, whether it is associated with developing chronicity in neck pain, a specific cut-off point to differentiate between these

**Table 1. Candidate prognostic factors.**

| Candidate prognostic factors | Measure | Range of the scale | Handling variables in statistical analysis |
|---|---|---|---|
| **Patients' characteristics** | | | |
| Sex | Self-report question | Male / Female | Dichotomous |
| Age | Self-report question | Age in years | Continuous |
| **Symptoms** | | | |
| Pain intensity at baseline | Numeric Pain Rating Scale (NPRS) "On a scale of 0 to 10, how much pain do you experience? Where 0 is no pain at all and 10 is the most imaginable pain" | 0–10 Higher scores indicate a higher degree of pain. | Continuous |
| Duration of neck pain | In weeks | Number of weeks | Continuous |
| Reported pain in different body regions | Self-report question: Do you also experience pain in other parts of your body? | Yes / No | Dichotomous |
| Accompanying headache | Self-report question: Have you experienced accompanying headache(s) since you have neck pain? | Yes / No/ I had headache(s) before the neck pain. | Dichotomous 1 = Yes 2 = No or I had headaches before the neck pain |
| Disability | Pain Disability Index (PDI) is a 7-item questionnaire to investigate the magnitude of self-reported pain-related disability. The PDI measures family/home responsibilities, recreation, social activity, occupation, sexual behavior, self-care, and life support [39] | 0–70 Higher scores indicate higher interference of pain with daily activity. | Continuous The sum score will be divided by the entered items. |
| **Work related factors** | | | |
| Happiness at work* | Self-report question: Can you indicate how happy you are with your current job? | Five point Likert scale.<br>• Totally unhappy<br>• Not happy<br>• Neutral<br>• Happy<br>• Totally happy | Dichotomous 1 = Happy (happy and totally happy) 2 = Not happy (totally unhappy, not happy and neutral) |
| Job satisfaction* | Self-report question: How much satisfaction do you get from your current job? | Five point Likert scale.<br>• Totally no satisfaction<br>• No satisfaction<br>• Neutral<br>• Satisfaction<br>• A lot of satisfaction | Dichotomous 1 = Satisfied (satisfaction and a lot of satisfaction) 2 = Not satisfied (totally no satisfaction, no satisfaction and neutral) |
| Potential to self-modify posture* | Self-report question: Are you able to change positions regularly during your work? | Five point Likert scale.<br>• Completely impossible<br>• Impossible<br>• Neutral<br>• Possible<br>• Completely possible | Dichotomous 1 = Possible 2 = Impossible |
| **General factors** | | | |
| Lifestyle Physical activity | Measured by the activity level according to the Dutch Healthy Exercise Norm | Dived into three categories:<br>1. I don't move 30 minutes any day a week of moderate intensity.<br>2. I'm exactly in between one and three<br>3. I am five days or more active per week | Dichotomous 1 = Achieving the Dutch Healthy Exercise Norm (category 3) 2 = Not achieving the Dutch Healthy Exercise Norm (category 1 and 2) |
| Smoking | Self-report question: Do you smoke? | Yes / No | Dichotomous |
| Alcohol | Self-report question: Do you drink alcohol? | Yes / No | Dichotomous |

*(Continued)*

**Table 1.** (Continued)

| Candidate prognostic factors | Measure | Range of the scale | Handling variables in statistical analysis |
|---|---|---|---|
| Length and weight | Self-report question:<br>What is your height?<br>What is your weight? | Body Mass Index (BMI): weight/ (length x length in meters) | Dichotomous |
| Sleep quality | Adjusted sleep quality question from the Neck Disability Index (NDI) and is subdivided in 4 domains; (1) wake up rested, (2) number of hours disturbed while sleeping, (3) fall asleep, and (4) personal experience sleep quality | (1) Yes / No<br>(2) 0–5 Higher scores indicate more hours disturbed while sleeping<br>(3) Yes / No difficulty falling asleep<br>(4) Yes / No personal experience difficulty sleeping or falling asleep | Dichotomous<br>1 = No negative experience with sleeping (No negative score on one of the four domains)<br>2 = Negative experience with sleeping (a positive score on one of the four domains) |
| **Psychological and behavior factors** | | | |
| Catastrophizing | Pain Catastrophizing Scale (PCS) short version is a 6-item questionnaire that assesses catastrophic thoughts or feelings associated with the experience of pain [40] | 0–24<br>Higher scores indicate more catastrophic thoughts | Continuous |
| Illness beliefs about recovery | Brief Illness Perception Questionnaire-Dutch language version (IPQ-DLV)<br>Two single questions:<br>How long do you think your neck pain will continue?<br>How concerned are you about your illness? | 0–10<br>0 a very short time– 10 forever<br>Higher scores indicate a maladaptive illness perception<br>0 not at all concerned– 10 extremely concerned<br>Higher scores indicate a maladaptive illness perception | Continuous |
| Treatment beliefs | Brief Illness Perception Questionnaire-DLV [41]<br>Single question:<br>How much do you think your treatment can help your neck pain? | 0–10<br>0 not at all—10 extremely helpful<br>A lower score indicates a maladaptive illness perception | Continuous |
| Depression | Depression Anxiety Stress Scale 21-item version (DASS-21) [42] | 0–21<br>Higher scores indicate a higher degree of depression | Continuous |
| Kinesiophobia | Tampa Scale for Kinesiophobia (TSK) 11-item version [43] | 11–44<br>Higher scores indicate a higher degree of kinesiophobia | Continuous |
| Distress | Depression Anxiety Stress Scale 21-item version (DASS-21) [42] | 0–21<br>Higher scores indicate a higher degree of stress | Continuous |
| Coping | Pain Coping Inventory (PCI) [44] is a 33-items questionnaire and is subdivided into six scales: pain transformation, distraction, reducing demands, retreating, worrying, and resting<br>Transforming the classification into an active (pain transformation, distraction and reducing demands) and passive coping strategy (retreating, worrying, resting) | Active coping = 12–48<br>Passive coping = 21–84 | Dichotomous |
| Illness beliefs about pain identity | Brief Illness Perception Questionnaire-DLV [41]<br>Single question:<br>How well do you feel you understand your illness? | 0–10<br>0 don't understand at all—10 understand very clearly<br>A lower score indicates a maladaptive illness perception | Continuous |
| Hypervigilance | Pain Vigilance Awareness Questionnaire (PVAQ) [45] | 0–80<br>Higher scores indicate a higher degree of vigilance | Continuous |
| Self-efficacy | Pain Self-efficacy Questionnaire 2-item version [46] | 0–12<br>Higher scores indicate a higher degree of self-efficacy | Continuous |
| **Remaining factors** | | | |

(*Continued*)

**Table 1.** (Continued)

| Candidate prognostic factors | Measure | Range of the scale | Handling variables in statistical analysis |
|---|---|---|---|
| Health care provider attitude* | Two vignettes consisting of 8 multiple choice questions and 4 open questions<br>The open questions focused on the history taking, examination and treatment strategy<br>The multiple-choice questions focus on the advice of the therapist with regard to categorizing of the complaint in type of seriousness, resumption of work and the implementation of daily activities | Biomedical<br>Biopsychosocial | Dichotomous |
| Therapeutic relation* | Self-report question<br>How much trust do you have in your healthcare provider/ physiotherapist? | 0–10<br>0 no trust at all– 10 very much confidence | Continuous |

*Candidate prognostic factors measured by an unvalidated measurement.

two levels does not exist. Therefore, our analyses will address catastrophizing and kinesiophobia as continuous factors.

The Depression Anxiety Stress Scale– 21 (DASS-21), recommended by Bijker et al. (2020) [49], is used to map the degree of stress and depression. The DASS-21 consists of 21 questions with three subscales: depression, anxiety, and stress. Each subscale consists of 7 questions with the answer ranging from 0 (not applicable at all or never applicable) to 3 (very definitely or mostly applicable) [42]. The internal consistency and test-retest reliability are sufficient for the DASS, and the convergent and divergent validity was supported [42].

The coping strategy of people with pain symptoms is measured through the Pain Coping Inventory List (PCI). This 33-item questionnaire reliably assesses six specific cognitive and behavioral strategies [44, 50]. The sensitivity and reproducibility of the PCI are acceptable [44]. Transforming the classification into an active or passive coping strategy is included in the content and construct validity. However, it has been validated in studies on chronic pain patients who experience physical complaints or (dis)function [50]. The items are scored using an ordinal measurement level from 1 (rarely) to 4 (very common).

The illness perceptions are measured with the Illness Perception Questionnaire–Dutch language version (IPQ-DLV) [41]. The IPQ-K is a cross-culturally adapted Dutch version of the Brief Illness Perception Questionnaire (BIPQ) [51]. Four out of eight questions from the IPQ-DLV were included in this study to measure patients' illness perceptions about recovery, treatment beliefs, and pain identity. The IPQ-DLV is an easy-to-understand questionnaire for patients and healthcare professionals. Each question represents a different disease perception with a different outcome measure. The items are scored using an ordinal measurement level from 0–10. The questionnaire has moderate to substantial reliability, acceptable face validity, and acceptable content validity [41]. The IPQ-K is an inventory questionnaire that can also be used evaluatively [41]. The reproducibility appeared to be moderate to good [51–53].

The degree of vigilance is assessed by the 16-item Pain Vigilance Awareness Questionnaire (PVAQ). Respondents are asked to think about their behavior in the past two weeks and indicate how often each item is a true reflection of their behavior or feelings. This questionnaire labeled two factors: "attention to pain" and "attention to changes in pain". The degree of vigilance is rated on a 6-point scale ranging from 0 (never) to 5 (always) [54, 55]. The PVAQ showed good validity, and internal consistency and fair test-retest reliability [45, 54].

The short version of the Pain Self-Efficacy Questionnaire (PSEQ-2) is a robust measure of pain self-efficacy and is recommended by Sleijser-Koehorst et al. (2019) [56]. It appears to be suitable for use in clinical and research settings [46].

**Remaining modifiable factors.** The therapist's orientation, biomedical (BM) or biopsychosocial (BPS), is assessed by asking the therapist to fill in two vignettes. Vignettes are a realistic simulation of case situations in daily practice to measure of diagnosis or evaluation by health care providers. It is a promising quality rating for estimating the clinical behavior of care providers and, if constructed correctly, is a valid measuring instrument [57, 58]. Vignette 1 (acute non-specific neck pain) consists of open questions (4) and multiple-choice questions (4). The open questions focus on the history taking, examination, and treatment strategy. The multiple-choice questions focus on the therapist's advice concerning the complaint in type and seriousness, resumption of work, and of daily activities. Vignette 2 (chronic non-specific neck pain) consist only of the multiple-choice questions (4). The vignettes used are based on standardized vignettes on low back pain [59].

In order to categorize the therapists (BM or BPS), the SCEBS method is used, covering Somatic, Psychological (Cognition, Emotion, and Behavior), and Social dimensions [60]. A therapist with a biomedical orientation believes in a biomedical model of disease, where disability and pain are a consequence of a specific pathology within the spinal tissues, and treatment is aimed at treating the pathology and alleviating the pain [59]. A therapist with a biopsychosocial orientation believes in a biopsychosocial model of disease in which pain does not have to be a consequence of tissue damage and can be influenced by social and psychological factors [59]. The open questions are scored on the emergence of the different dimensions of the SCEBS, whereby the somatic dimension scores as a more biomedical orientation, and the dimensions cognition, emotion, behavior, and social score as biopsychosocial orientation. The multiple-choice questions score as a more biomedical orientation if the therapist is more likely to rate for spinal pathology, recommend a delay in return to work and daily activity [61–63]. The scores are merged at the end to a sum score, which categorize a therapist as BM or BPS. Every therapist is categorized by two researchers individually; after scoring, there will be a consensus meeting between the two researchers. A third reviewer makes the final decision if consensus cannot not be reached.

Therapeutic relation is measured by a self-developed single question of which psychometric properties are unknown and was formulated based on a Delphi study [24].

## Sample size

To ensure the sample size is adequate in terms of the number of participants (*n*) and outcome events (*E*) relative to the number of predictor parameters (*p*) considered for inclusion, the minimum number of events per predictor parameter (EPP) is calculated recommended by Riley et al. (2019) [64]. To reduce the risk of overfitting and to ensure that the overall risk is estimated precisely, the following criteria need to be met: (1) small optimism in predictor effect estimated as defined by a global shrinkage factor of $\geq 0.85$, (2) small absolute difference of $\leq 0.05$ in the model's apparent and adjusted Nagelkerke's $R^2$, and (3) precise estimation of the overall risk of rate in the population or similarly, precise estimation of the model intercept when predictors are mean-centered [64]. The calculation of the expected value of the (Cox-Snell) R-squared of the new model is based on two included prognostic models and is estimated at $R^2 = 0.23$ [16, 65, 66]. The outcome events (*E*) are estimated at 45% based on a systematic review by dividing the included number of patients by the number of non-recovery of pain [16]. The number of included candidate predictor parameters for potential inclusion in the new model is based on a systematic review and a consensus study and is estimated at 26, of

which 4 are non-modifiable and 22 are potentially modifiable. The a priori sample size calculation for the prognostic model suggests to include a minimum of 598 participants.

## Statistical analysis methods and missing data

The statistical analysis is based on the 'Prognosis Research Strategy (PROGRESS) framework' type 3 research [26], in which the step-by-step plan will be roughly as follows:

- Analysis of cases with and without the development of the outcome events (whether or not they developed chronic pain, respectively) will be done to determine if there are significant differences. In case > 5% of incomplete records, data will be imputed. A multiple imputation strategy will be followed in case we assume data are at least missing at random. The number of imputations will be set to the percentage of incomplete records. Imputed values for continuous variables will be drawn using predictive mean matching. In case of evidence of data being MAR (or MCAR), the MAR assumption will be assessed by making a missingness indicator and testing whether incomplete patients differ from those that are incomplete.

- Identifying the independent predictive capacity of the candidate prognostic variables at baseline and the existence or non-existence of chronic pain measured at six weeks, three, and six months by univariable logistic regression analysis. These analyses will not be used to decide which prognostic factors will be included in the multivariable analyses.
  If the sample size, as calculated, turns out to be adequate, all variables will be include in the multivariable analyses.

- Multicollinearity between candidate predictors will be assessed using the variance inflation factor. In case the variance inflation factor exceeds 10, we will select which candidate predictor add to the modeling phase based on clinical expertise.

- The non-variable factors of age, gender, and duration of the pain will be included to strengthen our model. The discriminative ability of the prognostic model will be determined based on the Area Under the receiver operating characteristic Curve (AUC), calibration will be assessed using a calibration plot and formally tested using the Hosmer and Lemeshow goodness-of-fit test, and model fit will be quantified as Nagelkerke's $R^2$.

- Internal validation will be performed using bootstrap resampling to estimate the optimism-corrected AUC and to yield a measure of overfitting (i.e., the shrinkage factor). The shrinkage factor (a constant between 0 and 1) will be used to multiply the regression coefficient by. Generally, regression coefficients (and resulting predictions) are too extreme in case of overfitting, which is counteracted by the shrinking of regression coefficients.

## Discussion

This prospective cohort study will be the most extensive study in this field to determines prognostic factors for the chronification of acute- or subacute nonspecific idiopathic, non-traumatic neck pain in primary care physiotherapy. In contrast to most other prognostic research studies, this study has a biopsychosocial view and focuses specifically on potentially modifiable factors by a physiotherapist. By selecting patients in primary care physiotherapy practices, we assume that they will represent the usual population consulting the physiotherapist with neck pain. The results of this study will improve the understanding of prognostic and potential protective factors, which will help clinicians guide their clinical decision making, develop an individualized treatment approach, and predict chronic neck pain more accurately.

The candidate prognostic factors in this study are mostly modifiable. The non-modifiable factors of increasing age, sex, duration of neck pain, and reported pain in different body regions have a known prognostic value for neck pain patients [10, 15, 16, 67]. Therefore these will be included in the model development to strengthen the value of our prognostic model. However, their non-modifiable nature means that they have limited use in potential prevention strategies. To pursue the clinical applicability of the model, other potentially relevant and modifiable factors are selected for inclusion based on our systematic review and international Delphi study.

## Strengths and limitations

This study includes critical methodological features in order to minimize bias. These features include sampling a representative cohort from a physiotherapy setting with a high follow-up rate [68]. A new strategy for a representative sample size will be used. The rule-of-thumb events per variable (EPV) of $\geq 10$ is widely used in the medical literature as the lower limit for developing prediction models that predict a binary outcome. However, this generally accepted minimal sample size criterion has been found lenient when default stepwise predictor selection strategies develop prognostic models. Earlier critiques on EPV as a sample size criterion have identified its weak theoretical and empirical underpinning [69].

The new strategy to achieve an accurate sample size offers us space for 26 candidate prognostic factors in model development to avoid overfitting in our analyses. Because more candidate prognostic factors can lead to model overfitting in small data sets, spurious observed relationships can occur because of regression value distortion and an overestimating predictive performance [64, 70]. The 26 candidate prognostic factors permitted are selected based on our previous systematic review and Delphi study to include only relevant and potential important factors.

Although this study does not influence the therapy the participants receive, the given therapy may influence the outcome and the accuracy and transportability of the model to be developed [71]. The patients receive standard care based on the Dutch Physiotherapy Guideline for neck pain [34]. They may include therapy to modify our candidate prognostic factors and thereby have a risk-reducing effect on chronicity. In addition, there may also be a form of 'background treatment'; this could include any other treatment that an individual received during our prognostic study (e.g., psychological care) or changes an individual makes to their lifestyle [71]. We will have no information on this form of treatment during this study; however, it could influence the outcome. Nevertheless, we consider the impact on our study findings to be minimal, given (1) the heterogeneity of the factors to be modified, (2) the multiple modalities used by physiotherapists, and (3) the difference in physiotherapists' backgrounds. Thereby, we will report the physiotherapy treatment the patient received and discuss the possible impact on our study findings (TRIPOD 5C) but do not include the different treatments as a predictor in our model. Moreover, the current setting does reflect clinical practice as it is. This heterogeneity is likely to remain even after implementing of a well-performing model.

## Clinical message and future directions

This study protocol describes only the first phase of prognostic model research; model development (including internal validation). Our model should be externally validated using data from another dataset to assess the generalizability of our prognostic model [72]. Thereafter, investigations of impact on decision-making and patient outcomes have to be done to measure our study's clinical relevance and impact.

## Acknowledgments

We want to thank all the physiotherapists who will include and follow their patients for six months. We would also wish to extend our special thanks to all the patients who will participate in this research.

We want to thank all reviewers, Assoc. Prof. Dr. Farnaza Ariffin, Prof. Alice Kongsted, Prof. Saud M. Al-Obaidi Phd PT, and Ass. Prof. Yousef Alshere, for taking the time and effort necessary to review the manuscript. We sincerely appreciate all valuable comments and suggestions which helped us improve the manuscript's quality.

## Author Contributions

**Conceptualization:** Martine J. Verwoerd, Harriet Wittink, Francois Maissan, Rob J. E. M. Smeets.

**Funding acquisition:** Martine J. Verwoerd, Harriet Wittink.

**Methodology:** Martine J. Verwoerd, Harriet Wittink, Sander M. J. van Kuijk, Rob J. E. M. Smeets.

**Writing – original draft:** Martine J. Verwoerd.

**Writing – review & editing:** Martine J. Verwoerd, Harriet Wittink, Francois Maissan, Sander M. J. van Kuijk, Rob J. E. M. Smeets.

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
