## [Decision Letter · Decision Letter 0]

18 Oct 2022

PONE-D-22-19562

A study protocol for the validation of a prognostic model with an emphasis on modifiable factors to predict chronic pain after a new episode of acute- or subacute nonspecific idiopathic, non-traumatic neck pain presenting in primary care.

PLOS ONE

Dear Dr. Verwoerd,

Thank you for submitting your manuscript to PLOS ONE. After careful consideration, we feel that it has merit but does not fully meet PLOS ONE’s publication criteria as it currently stands. Therefore, we invite you to submit a revised version of the manuscript that addresses the points raised during the review process.

Your manuscript needs more clarification related to methods and statistical analysis based on reviewers' suggestions. Please, revise accordingly.

We look forward to receiving your revised manuscript.

Kind regards,

Aqeel M Alenazi

Academic Editor

PLOS ONE

Journal Requirements:

Reviewers' comments:

Reviewer's Responses to Questions

**Comments to the Author**

1. Does the manuscript provide a valid rationale for the proposed study, with clearly identified and justified research questions?

Reviewer #1: Yes

Reviewer #2: Partly

Reviewer #3: Yes

Reviewer #4: Yes

2. Is the protocol technically sound and planned in a manner that will lead to a meaningful outcome and allow testing the stated hypotheses?

Reviewer #1: Yes

Reviewer #2: Yes

Reviewer #3: Yes

Reviewer #4: Yes

3. Is the methodology feasible and described in sufficient detail to allow the work to be replicable?

Reviewer #1: Yes

Reviewer #2: Yes

Reviewer #3: Yes

Reviewer #4: Yes

4. Have the authors described where all data underlying the findings will be made available when the study is complete?

Reviewer #1: No

Reviewer #2: No

Reviewer #3: Yes

Reviewer #4: No

5. Is the manuscript presented in an intelligible fashion and written in standard English?

Reviewer #1: Yes

Reviewer #2: Yes

Reviewer #3: Yes

Reviewer #4: Yes

6. Review Comments to the Author

You may also provide optional suggestions and comments to authors that they might find helpful in planning their study.

Reviewer #1: Dear author, here are my comments

1. Title: acceptable and represents the study objective

2. Introduction: acceptable

3. Objective in the manuscript protocol MUST be the same as the objective in the abstract. Suggest to focus on the modifiable. (please correct the manuscript objective)

4. the inclusion and exclusion criteria are not coherent. Also, the arrangement of the inclusion and exclusion criteria are confusing. Start with inclusion criteria: Patients are 18 years and older; those with a new presentation of neck pain not more than 12 weeks upon onset; no (3); no (4) then for exclusion criteria, What does generalized pain mean? please specify is it generalized all over body pain? and also please arrange according to general and specific exclusions.

5. For the outcomes, there are a lot of scales that you want to use. Please provide a table of the main outcome and all the other outcome measures which includes the scales you are using.

In general, this is a comprehensive study and will provide valuable information for the prognosis of chronic pain among those presenting with acute / subacute neck pain. However, the manuscript protocol needs some adjustment and correction to make it more succinct and understandable to the readers.

Thank you.

Reviewer #2: Thanks for the opportunity to review this protocol. As the data collection is almost finished, I have focused on the reporting of the protocol rather than the design. Generally, I find it clearly reported and the study well thought through.

A few comments for the author to consider.

The study is described as PROGRESS Type 3, consider adding that it is a type 1b according to TRIPOD.

Candidate prognostic factors:

As you are clearly aware, most developed prediction models end out being of little value, because they do not perform well outside of the development sample. Knowing that, I am not sure why you did not use an existing model as starting point and then updated with new predictors if relevant. Please consider addressing that in the introduction or discussion.

For some of the factors listed in Table 1, which are not presented as “unvalidated”, there is no reference provided to the validation of the measure. Please make sure that you either provide supporting information or state it is an unvalidated measure.

You argue for the use of age, gender, and symptom duration in addition to potentially modifiable factors. I agree with that decision but think the same argument would apply to for instance socioeconomic factors. Consider addressing why you see these differently than age, gender, and duration. I follow your argument why it is most helpful to identify modifiable factors, but if unmodifiable are strong determinants of outcome that is important for informing management strategies as well.

Outcome:

Please report the wording of the pain intensity measure (current pain, average pain last x weeks, or??) Note that you have used NPRS for ”Numeric pain rating scale” and for “Neck pain rating scale”.

Analysis plan:

The description of how you will determine if missing data is missing at random is not really clear to me. Consider if it needs rewording.

It is not clear why you perform the univariable analyses. I think it is fine to present these to inform readers about their relationship with the outcome. Perhaps state directly that univariable analyses are not used to decide which prognostic factors will go into the multivariable model.

I am not a fan of considering the inclusion of age and gender an “adjusted” prediction model. I think that tends to mix concepts between prediction and causation. In prediction models, all included variables potentially contribute to predictive performance. You may consider investigating if modifiable predictors interact with age or gender if the sample size allows for that.

Your plan for model reduction is somewhat hidden under internal validation. It would be useful to briefly describe how shrinkage will be used to remove potential predictors. How close to zero should the coefficient by after shrinkage for the factor to be removed from the model?

I recommend that you report how you plan to compare the performance to existing prediction models, including which models you will compare to. How will you determine if it is worthwhile moving on to external validation?

Reviewer #3: PLOS ONE PONE-D-22-19562

A study protocol for the validation of a prognostic model with an emphasis on modifiable factors to predict chronic pain after a new episode of acute- or subacute nonspecific idiopathic, non-traumatic neck pain presenting in primary care.

I highly commend the authors for their high quality work and the potential prospect impact on health care and clinical decision making in the care of neck pain and prediction of potential factors that may lead to chronicity, in deed this is a great valuable research efforts.

However, the following is my humble criticism

Objectives

You wrote ;

“The primary objective of this study is to identify which modifiable and non-modifiable factors are independent predictors of the development of chronic pain in patients with acute- or subacute nonspecific idiopathic, non-traumatic neck pain, and secondly, to combine these to develop and internally validate a prognostic prediction model”.

*With regards to the objectives I fully I agree with the authors . Their objectives are legitimate and rational , and these kind of studies is needed to improve the clinical work of physiotherapist.

However, I should point out that because there is a synchronized positive relationship between pain, and fear related illness behaviors in acute sage most of the psychological factors cannot be modifiable at acute stage because it is natural response to pain. However, this relationship may vanish or persist in most people following subacute stage and may be reinforced in others due to selected environmental biopsychosocial factors as you indicated. The role of the psychological factors mainly fear of/ anticipation of pain and catastrophizing continue to develop and reinforce the pain and related behavior beyond the subacute stage, so the argument that may arise her is whether the emphasis should be directed to screen for the most frequent reinforcers found in the chronic stage and then to be used later in acute or subacute stage as prognosis predictors so that the therapist screen to identify and modify.

Study Settings

You have large Number of physiotherapist included in the study this may have advantage or disadvantage specifically, related to these individual knowledge and understanding of the biopsychological factors, and skills in screening, or treatment implementation of these knowledge in their workplace this may need to elaborated on don you agree .

Participant

“The patients are at least 18 years or older”

Neck pain is common among adults, although it can occur at any age. Are you a wear of the Global Burden of Diseases 2017 study, it demonstrated that the point prevalence of neck pain peaked during the middle ages with the highest burdens in people age 45–49 and 50–54 for men and women, respectively.

Kazeminasab, S., Nejadghaderi, S.A., Amiri, P. et al. Neck pain: global epidemiology, trends and risk factors. BMC Musculoskelet Disord 23, 26 (2022). https://doi.org/10.1186/s12891-021-04957-4

Therefore, including young participant starting from age 18 years may not reflect the day today patient volume in any physical therapy clinical setting. In addition, young individuals are motivated, highly active and may have different biopsychosocial factors than the adults or senior individual.

You may need to reflect on the young age pain behavior psychology if you have a huge number of young individuals or eliminate young individuals as a sub population in your study.

Item 4 “If the patient has had neck pain before, the patients must be relatively free from symptoms for at least three months”

My question to you is how would you assume that this patient has no learning effect from his previous episodes pain related fear experience and behavior , I though you needed an individuals with totally knew pain experience to be able to demonstrate a good predicting tool.

I have reviewed the Delphi survey and your systematic survey I think I perceived that unhealthy life style and physical inactivity were considered separate variables,

I also did not find any mentioned to somatization in your factors, this was mentioned earlier by the Delphi survey could you define it and further elaborate why it was not included

• Her is another confusing problem for the prognostic tool you are about to produce and please allow me to elaborate and explain my worries and its all centered around Kinesiophobia.

Kinesiophobia defined as “an excessive, irrational, and debilitating fear of physical movement and activity resulting from a feeling of vulnerability due to painful injury or reinjury”

It is by itself a central factor in the process of pain developing from acute to chronic stages, The Cognitive Fear Avoidance Model describes that very clearly , it also linked it strongly to catastrophizing, re/injury, avoidance behaviour, and on the long run inactivity, disability, depression, low self-esteem and increasing the risk for a wide range of health problems, functional decline and premature death.

My question is

You have included so many independent factors that have high potential prognostic ability alone, so it become very difficult and confusing to understand the weighting of theses variables to the final prediction scores on your prognostic tool while in reality each one of which is so representative and highly weighted predictor itself such as kinesiophobia , depression, anxiety, low self-esteem. Need to elaborate a bit more on this.

Discussion

Line 338-350 I found the following statement very confusing and not helpful to justify the objectives of the study

338. The given therapy may influence the outcome and the accuracy and transportability of the model to be developed.

340. The patients receive standard care based on the Dutch Physiotherapy Guideline for neck pain.

344-345. We will have no information on this form of treatment during this study; however, it could influence the outcome.

You wrote “Nevertheless, we consider the impact on our study findings to be minimal, given ……..”

In their conclusion Carroll LJ, Hogg-Johnson S, van der Velde G, Haldeman S, Holm LW, Carragee EJ, Hurwitz EL, Côté P, Nordin M, Peloso PM, Guzman J, Cassidy JD. Course and Prognostic Factors for Neck Pain in the General Population: Results of the Bone and Joint Decade 2000 –2010 Task Force on Neck Pain and Its Associated Disorders. Eur Spine J. 2008 Apr;17(Suppl 1):75–82. doi: 10.1007/s00586-008-0627-8. Epub 2008 Feb 29. PMCID: PMC2271093. “General exercise was not prognostic of better outcome; however, several psychosocial factors were prognostic of outcome”.

In their discussion they also have specifically stated that “ Prognosis may also depend on whether or not the exercises themselves were designed to impact the

neck and shoulder areas”.

My question

since the author of the above study concluded only on general exercises, but have raised the potential value of specific designed exercise then this raise the concerns about the type of approach used in these multicenter and by the physiotherapist skills and preferences involved in this study .

The benefit and impact of specific approaches as the directed self treatment exercise for the neck used by well trained therapist such as Maitland, McKenzie, or Shacklock M neurodynamic and others on the outcomes has ben well documented such techniques has prognostic value against psychological factors and chronicity.

For example directional preference exercise by the McKenzie approach help the patient face his one fear of painful movement direction while rehearsing in this direction which help centralized improve and prevent the next episodes of pain and fast return to work.

Line 348-350

You wrote

“Thereby, we will report the physiotherapy treatment

the patient received and discuss the possible impact on our study findings (TRIPOD 5C) but do not include the different treatments as a predictor in our model” .

I think you may need to reconsider these statement in light of the variation of approaches among theses centers to reveal the impact of these specific neck exercise approach on modifying the psychological factors

Reviewer #4: I would like to thank the authors for their time and efforts in drafting this protocol study entitled, “A study protocol for the validation of a prognostic model with an emphasis on modifiable factors to predict chronic pain after a new episode of acute- or subacute nonspecific idiopathic, non-traumatic neck pain presenting in primary care”.

I found it to be of interest and very well written. However, the authors may consider clarifying if the participants in their study will be asked to report pain intensity “in the last 24 hours” or “an average pain intensity” because recall of pain may vary.

I strongly suggest that the authors include the operational definitions for acute and subacute neck pain knowing that determining whether a patient fits into the acute or subacute time-period is not always as precise as we would like to think. In addition, I recommend including the participants’ number of previous episodes of neck pain as a factor in the analysis because number of previous episodes is associated with subsequent outcome in musculoskeletal pain symptom.

Although the time-period “0–4 weeks of symptoms” is well-known definition of “acute” neck pain, during analysis, I suggest (if applicable) exploring the relationship between participants prognostic factors at baseline and outcome for a subgroup of patients with acute neck pain with shorter period (0–2 weeks) “high-acute” and (2-4) “mid-acute” of symptoms.

Lastly, one formatting note: I believe the full stop should be placed after the in-text citation.

Thank You!

7. PLOS authors have the option to publish the peer review history of their article (what does this mean?). If published, this will include your full peer review and any attached files.

Reviewer #1: **Yes: **Associate Professor Dr Farnaza Ariffin

Reviewer #2: **Yes: **Alice Kongsted

Reviewer #3: No

Reviewer #4: **Yes: **Yousef M. Alshehre, PhD

---

## [Author Response · Author response to Decision Letter 0]

14 Dec 2022

Reviewer 1

In general, this is a comprehensive study and will provide valuable information for the prognosis of chronic pain among those presenting with acute / subacute neck pain. However, the manuscript protocol needs some adjustment and correction to make it more succinct and understandable to the readers.

Comment 1: Objective in the manuscript protocol MUST be the same as the objective in the abstract. Suggest to focus on the modifiable. (please correct the manuscript objective)

Thank you for making us aware of this. We changed the objective in the abstract of our manuscript on page 2, lines 22 - 25. 

The primary objective of this study is to identify which modifiable and non-modifiable factors are independent predictors of the development of chronic pain in patients with acute- or subacute nonspecific idiopathic, non-traumatic neck pain, and secondly, to combine these to develop and internally validate a prognostic prediction model. 

Comment 2: the inclusion and exclusion criteria are not coherent. Also, the arrangement of the inclusion and exclusion criteria are confusing. Start with inclusion criteria: Patients are 18 years and older; those with a new presentation of neck pain not more than 12 weeks upon onset; no (3); no (4) Ithen for exclusion criteria, What does generalized pain mean? please specify is it generalized all over body pain? and also please arrange according to general and specific exclusions.

Thank you for these suggestions. We have adjusted the order and changed the wording of the inclusion criteria. We now realize that ‘generalized pain’ is not an internationally accepted or precise term. With our exclusion criterion ‘generalized pain’ we meant the widespread pain as described in the ICD 11: chronic primary pain; diffuse musculoskeletal pain in at least 4 of 5 body regions and at least 3 or more body quadrants (as defined by upper–lower/left–right side of the body) and axial skeleton (neck, back, chest, and abdomen). 

We changed this exclusion criterion in:

(page 7, lines 137-139) Widespread pain (ICD 11); diffuse musculoskeletal pain in at least 4 of 5 body regions and at least 3 or more body quadrants (as defined by upper-lower / left-right side of the body) and axial skeleton (neck, back, chest, and abdomen).

We also describe the fourth exclusion criterion more specifically. We changed “pain not caused by a musculoskeletal origin” in: 

(page 8, lines 140-141) “Pain not caused by a musculoskeletal origin (not located in the muscles, bones, joints, or tendons)” (34).

In addition, we also categorized the exclusion criteria into general and specific (page 8-9). 

Comment 3: For the outcomes, there are a lot of scales that you want to use. Please provide a table of the main outcome and all the other outcome measures which includes the scales you are using.

We hope that we have interpreted this comment well, but we think we already provided a Table with that information, please see Table 1 (page 10-12) In this Table Candidate Prognostic factors, we present the different Candidate prognostic factors, measurement, the range of the scales, and how we will handle variables in our statistical analysis.

Reviewer 2

Thanks for the opportunity to review this protocol. As the data collection is almost finished, I have focused on the reporting of the protocol rather than the design. Generally, I find it clearly reported and the study well thought through.

A few comments for the author to consider.

Comment 1: The study is described as PROGRESS Type 3, consider adding that it is a type 1b according to TRIPOD.

Thank you for this relevant suggestion. We now add that we performing a type 1b study according to TRIPOD. 

(page 6, lines 98-102) The present study is a prospective cohort study of prognostic factors informed by the PROGRESS framework and TRIPOD statement type 1b and specific recommendations for statistical approaches to Type 3 prognostic model research (27,28).

Comment 2: Candidate prognostic factors: As you are clearly aware, most developed prediction models end out being of little value, because they do not perform well outside of the development sample. Knowing that, I am not sure why you did not use an existing model as starting point and then updated with new predictors if relevant. Please consider addressing that in the introduction or discussion.

We selected several candidate prognostic factors presented in previously published prognostic models. We found the following candidate prognostic factors in our systematic review: higher age, concomitant LBP of headache, and a previous episode of neck pain. Although the quality of the developed models was low, the studies were graded as low to very low and were thereby not applicable as a specific starting point. 

We do describe that we included several prognostic factors found in our systematic review in our method section under candidate prognostic factors and in our discussion. As this may not have been very clear in our manuscript, we make this more specific in the revised manuscript: 

(page 9, lines 170-172) “The candidate prognostic factors are based on our previous systematic review and Delphi study. From the systematic review, we included the variables significantly predictive of pain chronification or non-recovery. Furthermore, we included the variable with a consensus of >70% in the first round of our Delphi study.”

(page 20, lines 340-346) “The candidate prognostic factors in this study are mostly modifiable. The non-modifiable factors of increasing age, sex, duration of neck pain and reported pain in different body regions have a known prognostic value for neck pain patients(10,15,23,67). Therefore these will be included in the model development to strengthen the value of our prognostic model.”

Moreover, we agree that our considerations are not very well introduced. We do now in our introduction:

(page 4, lines 71-74) At the present time the existing literature on prognostic models shows a low performance in predicting chronicity or recovery from neck pain(15,16). It is therefore not applicable as a starting point for a new prognostic study. A limitation and possible explanation of this low performance is the inclusion of a too-heterogeneous group of neck pain patients. 

(page 5, lines 82-83) Only clinically modifiable factors have the potential to change patient outcome and are therefore recommended to be included in prognostic research(16,24). However, to strengthen a prognostic model, it can be relevant to include some non-modifiable factors. Based on a recent consensus study of potential modifiable prognostic factors, including psychosocial factors in prognostic research for chronification is relevant(25). 

Comment 3: For some of the factors listed in Table 1, which are not presented as “unvalidated”, there is no reference provided to the validation of the measure. Please make sure that you either provide supporting information or state it is an unvalidated measure.

Unfortunately, there were no suitable validated measurements for some of our candidate prognostic factors (work-related factors, therapeutic relations, and physiotherapists' attitudes). We describe this in our detailed description of the measurements. However, we agree that more attention must be given to the fact that we measure some factors with unvalidated questions or measurements in Table 1. 

We now do that by adding an asterisk to these factors and adding text underneath the Table: 

(page 12, line 177) *Candidate prognostic factors measured by an unvalidated measurement. 

Comment 4: You argue for the use of age, gender, and symptom duration in addition to potentially modifiable factors. I agree with that decision but think the same argument would apply to for instance socioeconomic factors. Consider addressing why you see these differently than age, gender, and duration. I follow your argument why it is most helpful to identify modifiable factors, but if unmodifiable are strong determinants of outcome that is important for informing management strategies as well.

Thank you for this suggestion. The candidate non-modifiable prognostic factors, age, gender, and symptom durations, all showed promising prognostic capacity in previous studies. Socioeconomic factors, such as social class, marital status, children, level of education, and employment status, were also included in previous studies as potential prognostic factors for pain or perceived non-recovery. Although some of these factors were positively associated with higher pain intensity or perceived non-recovery in the univariate analyses after 6 or 12 months, only employment status remained in one multivariable prognostic model. (Verwoerd, 2019) 

However, the impact of different socioeconomic factors on the prevalence, prognosis, and transition of pain is shown in several studies (Larsson B. et al., 2012, Jackson et al., 2015, Palmlöf L et al., 2012). Therefore, we do understand this critical point and concerns. However, we had to make choices in the candidate prognostic factors that we could include in this cohort study concerning the number of possible sample sizes to be achieved. Since too many candidate predictor factors towards a sample size is associated with overfitting. Therefore, we did not include these factors as candidate prognostic factors.

We included variables (1) work status and (2) education level in our patients’ questionnaire. See also our response to comment 6. We use these variables to describe the characteristics of our population, so professionals can decide whether our model can be used within their population.

Comment 5: Outcome: Please report the wording of the pain intensity measure (current pain, average pain last x weeks, or??) Note that you have used NPRS for ”Numeric pain rating scale” and for “Neck pain rating scale”.

We asked the patients to rate their current pain. Our exact question in our questionnaire is: “on a scale of 0 to 10, how much pain do you experience? 0 is no pain at all and 10 is the most imaginable pain.”

We added this specific question to table 1. 

Furthermore, in our description of the candidate prognostic factors under symptoms, the symptoms are current pain intensity. (page 13, line 179) 

We changed “Neck pain rating scale” in the “Numeric Pain Rating Scale”. Thank you for making us aware of this. 

Comment 6: Analysis plan: The description of how you will determine if missing data is missing at random is not really clear to me. Consider if it needs rewording.

It is not clear why you perform the univariable analyses. I think it is fine to present these to inform readers about their relationship with the outcome. Perhaps state directly that univariable analyses are not used to decide which prognostic factors will go into the multivariable model.

Thank you for these comments. We need to describe that we will not use the univariable analyses to decide which prognostic factors will be included in our multivariable analyses. In addition, it may have needed to be clarified why we are performing the univariate analysis. We do this now in our method section, under our statistical analysis methods and missing data (page 18; lines 310-315): 

Identifying the independent predictive capacity of the candidate prognostic variables at baseline and the existence or non-existence of chronic pain measured at six weeks, three, and six months by univariable logistic regression analysis. These analyses will not be used to decide which prognostic factors will be included in the multivariable analyses;

Regarding the missing data, we now write (page 18; lines 302-309): 

- Analysis of cases with and without the development of the outcome events (whether or not they developed chronic pain, respectively) will be done to determine if there are significant differences. In case > 5% of incomplete records, data will be imputed. A multiple imputation strategy will be followed in case we assume data are at least missing at random. The number of imputations will be set to the percentage of incomplete records. Imputed values for continuous variables will be drawn using predictive mean matching. In case of evidence of data being MAR (or MCAR), the MAR assumption will be assessed by making a missingness indicator and testing whether incomplete patients differ from those that are incomplete.

Comment 7: I am not a fan of considering the inclusion of age and gender an “adjusted” prediction model. I think that tends to mix concepts between prediction and causation. In prediction models, all included variables potentially contribute to predictive performance. You may consider investigating if modifiable predictors interact with age or gender if the sample size allows for that.

We agree that including age and gender as an adjusted factor does not seem right. It is also not sufficiently consistent with our statement that we use these factors as strengthening factors for our model development (page 20, line 340-343):

The non-modifiable factors of increasing age, sex, duration of neck pain and reported pain in different body regions have a known prognostic value for neck pain patients(10,15,23,67), therefore these will be included in the model development to strengthen the value of our prognostic model.

Moreover, in our revised version of our manuscript, we write now in our introduction (page 5, lines 82-83):

However, to strengthen a prognostic model, it can be relevant to include some non-modifiable factors.

We now write in our ‘Statistical analysis methods and missing data’ section (page 18, lines 319-320):

- The non-variable factors of age, gender, and duration of the pain will be included to strengthen the performance of our model. The discriminative ability of the prognostic model will be determined based on the Area Under the receiver operating characteristic Curve (AUC), calibration will be assessed using a calibration plot and formally tested using the Hosmer and Lemeshow goodness-of-fit test, and model fit will be quantified as Nagelkerke’s R2; 

Comment 8: Your plan for model reduction is somewhat hidden under internal validation. It would be useful to briefly describe how shrinkage will be used to remove potential predictors. How close to zero should the coefficient by after shrinkage for the factor to be removed from the model?

Thank you for this comment. By shrinking coefficients (slightly) towards 0, the effect of overfitting is counteracted. The internal validation step will not be used to select predictors for the prediction model, as e.g., the Lasso would do. Contrary to parameterwise shrinkage methods, the bootstrap yields a global shrinkage factor that is applied to all regression coefficients. 

See our changes at page 19, lines 324-328:

- Internal validation will be performed using bootstrap resampling to estimate the optimism-corrected AUC and to yield a measure of overfitting (i.e., the shrinkage factor). 

The shrinkage factor (a constant between 0 and 1) will be used to multiply the regression coefficient by. Generally, regression coefficients (and resulting predictions) are too extreme in case of overfitting, which is counteracted by the shrinking of regression coefficients. 

Comment 9: I recommend that you report how you plan to compare the performance to existing prediction models, including which models you will compare to. How will you determine if it is worthwhile moving on to external validation? 

Thank you for raising these critical points. Our systematic review showed that the previously developed models for this specific group of patients are based on studies with a high risk of bias and without good predictive performance. So, comparing a newly developed model with these previously developed models is not valid. In addition, we also focus on other variables, in particular modifiable factors, compared to the previously developed models.

Concerning pre-determining whether external validation is relevant, this cannot be determined in advance. A small discriminative ability may be relevant in a population with a high incidence of (chronic)neck pain and may thereby be socially relevant. In addition, if we find evidence that some of the included modifiable factors have a high predictive power, it will be relevant to clinical practice; since a physiotherapist might modify these factors. We would like to refrain from including this dilemma in our protocol, but will undoubtedly include this in the discussion of our results manuscript.

 

Reviewer 3

A study protocol for the validation of a prognostic model with an emphasis on modifiable factors to predict chronic pain after a new episode of acute- or subacute nonspecific idiopathic, non-traumatic neck pain presenting in primary care.

I highly commend the authors for their high quality work and the potential prospect impact on health care and clinical decision making in the care of neck pain and prediction of potential factors that may lead to chronicity, indeed this is a great valuable research efforts. However, the following is my humble criticism

Comment 1: Objectives You wrote; “The primary objective of this study is to identify which modifiable and non-modifiable factors are independent predictors of the development of chronic pain in patients with acute- or subacute nonspecific idiopathic, non-traumatic neck pain, and secondly, to combine these to develop and internally validate a prognostic prediction model”. *With regards to the objectives I fully I agree with the authors . Their objectives are legitimate and rational , and these kind of studies is needed to improve the clinical work of physiotherapist.

However, I should point out that because there is a synchronized positive relationship between pain, and fear related illness behaviors in acute sage most of the psychological factors cannot be modifiable at acute stage because it is natural response to pain. However, this relationship may vanish or persist in most people following subacute stage and may be reinforced in others due to selected environmental biopsychosocial factors as you indicated. The role of the psychological factors mainly fear of/ anticipation of pain and catastrophizing continue to develop and reinforce the pain and related behavior beyond the subacute stage, so the argument that may arise her is whether the emphasis should be directed to screen for the most frequent reinforcers found in the chronic stage and then to be used later in acute or subacute stage as prognosis predictors so that the therapist screen to identify and modify.

Thank you for this comment. We agree that some of our included factors can be a natural response to pain, especially kinesiophobia and catastrophizing. A low level of kinesiophobia or catastrophizing can be a healthy and beneficial response to pain, giving the body some time and rest to recover. However, the level of some factors, for instance, kinesiophobia and catastrophizing, differ between patients. It is known that an excessively negative orientation toward pain (pain catastrophizing and fear of movement (kinesiophobia) is essential in the etiology of chronic low back pain (Picavet et al, 2002). That is also why we decided to handle the level of kinesiophobia and catastrophizing as continuous factors in our analyses; a specific cut-off point is not known in when it is a healthy or an excessive response that can be associated with or is prognostic for chronicity.

Concerning the potential limited possibility to modify psychological factors in a (sub)acute phase; data suggest that behavioral interventions are associated with concurrent changed in catastrophizing and pain intensity in acute and chronic pain states (George SZ et al., 2008; Phillip J Quartana et al., 2009). Nevertheless, it is difficult to determine whether the stronger magnitude of the relationship for state versus trait pain catastrophizing is not attributable in large part to confounding with the pain experience itself (Phillip J Quartana et al., 2009). However, since we develop a prognostic model for a physical therapist in the acute stage, measuring these psychologic factors in the (sub)acute phase of neck pain is the only option.

Thanks to your critical comment, and since we explicitly considered this, we have clarified this in our revised manuscript (page 14, lines 217-221):

In a (sub)acute state of pain, a response such as fear of movement or negative orientation toward pain could exist. However, it is not known when this response is a beneficial level of adaptation or an excessive response to (sub)acute pain. Furthermore, whether it is associated with developing chronicity in neck pain, a specific cut-off point to differentiate between these two levels does not exist. Therefore, our analyses will address catastrophizing and kinesiophobia as continuous factors.

Comment 2: Study Settings. You have large Number of physiotherapist included in the study this may have advantage or disadvantage specifically, related to these individual knowledge and understanding of the biopsychological factors, and skills in screening, or treatment implementation of these knowledge in their workplace this may need to elaborated on dont you agree .

We agree that there will be a difference in understanding of the biopsychological factors and skills in screening between our included physiotherapists. However, this fact does not directly influence our study because screening the candidate (psychological)prognostic factors will be done with patient-reported questionnaires. 

There will also be a difference in the treatment of our included patients. We cover that by including the attitude of the physiotherapists as a candidate prognostic factor. 

Suppose our prognostic model will be sufficient and physiotherapists’ attitudes are one of the prognostic factors, next we could try to change these attitudes in practice. In addition, that may also change practice behavior (e.g., biopsychosocial skills) (Verwoerd, 2021).

Comment 3: Participant. “The patients are at least 18 years or older”

Neck pain is common among adults, although it can occur at any age. Are you awear of the Global Burden of Diseases 2017 study, it demonstrated that the point prevalence of neck pain peaked during the middle ages with the highest burdens in people age 45–49 and 50–54 for men and women, respectively. Kazeminasab, S., Nejadghaderi, S.A., Amiri, P. et al. Neck pain: global epidemiology, trends and risk factors. BMC Musculoskelet Disord 23, 26 (2022). https://doi.org/10.1186/s12891-021-04957-4

Therefore, including young participant starting from age 18 years may not reflect the day today patient volume in any physical therapy clinical setting. In addition, young individuals are motivated, highly active and may have different biopsychosocial factors than the adults or senior individual. You may need to reflect on the young age pain behavior psychology if you have a huge number of young individuals or eliminate young individuals as a sub population in your study.

We are aware of the highest burdens of neck pain in different age groups for men and women. However, these numbers are not based on the visiting patients at physiotherapy primary care or patients with non-specific acute- or subacute neck pain. There is no description of these age groups' neck pain; it can be mainly chronic or specific neck pain. Therefore, we will not only include or make subgroups of patients in a particular age group beforehand. Please note that all consecutive patients will be included; hence, we expect to include a very broad range of ages in our study. Moreover, age has been selected as a candidate prognostic factor and may be included if shown to be an important predictor of the outcome.

In addition, the model we will develop will be made for primary care and therefore has to be generalizable to that setting. That is why we must include the entire group with acute and subacute complaints that enter primary care in the Netherlands. This will make the model applicable and clinically relevant in primary care. However, it is an interesting point. We will measure the central tendency in our descriptive statistics. Based on these findings, we can compare the incidence of our patients with neck pain and the one in the global burden of diseases study.

Nevertheless, if age and related factors (high activity, biopsychosocial factors, etc.) significantly impact prognosis, we will find that in our univariate and multivariate analyses.

Comment 4: “If the patient has had neck pain before, the patients must be relatively free from symptoms for at least three months”

My question to you is how would you assume that this patient has no learning effect from his previous episodes pain related fear experience and behavior , I though you needed an individuals with totally knew pain experience to be able to demonstrate a good predicting tool.

We are aware that some patients may have “learned” from their previous pain episodes. However, this was a pragmatic choice as large sample size is needed to develop prognostic models, and we foresaw difficulty collecting “fresh” subjects. We ask the included patients if they had a previous pain episode, and we will present that in the characteristics of our population. We will report on this in the discussion of the results paper as a potential limitation of our study.

Comment 5: I have reviewed the Delphi survey and your systematic survey I think I perceived that unhealthy life style and physical inactivity were considered separate variables, I also did not find any mentioned to somatization in your factors, this was mentioned earlier by the Delphi survey could you define it and further elaborate why it was not included. 

Thank you for looking critically at the various studies we conducted prior to this cohort study and the choices we made based on these studies.

In our Delphi study, physical inactivity emerges as a separate variable. However, physical inactivity is one of the domains of a lifestyle. Therefore, we have placed several variables under lifestyle; alcohol use, smoking, BMI, and physical inactivity. We should have made that clear in Table 1. We have adjusted this, so it is clear that these are all different candidate prognostic factors included in our analysis (see Table 1, page 12). 

You are right about the variable somatization. Based on our statement that we would include all variables above >70% consensus in our cohort study, this should be included. However, we had to make choices in the number of variables we could include in our study. Achieving a sample size large enough for the number of variables in such a specific group of patients is difficult within the time we have. We included the variables that scored above >70% in the first round of the Delphi. We did this because the degree of bias in the second round of our Delphi study may be high. We have also described this in the discussion of our Delphi study. 

We need to describe this clearly in our protocol. We now do this in our method section under 'Candidate prognostic factors' (page 9, line 170-172): 

From the systematic review, we included the variables significantly predictive of pain chronification. Furthermore, we included the variable with a consensus of >70% in the first round of our Delphi study.

However, the fact that somatization does emerge as a prognostic or risk factor in various studies could be a reason to include it (Farideh Sadeghian et al., 2012; C K Jørgensen et al. 2000; Mehling WE et al. 2015; Markus Melloh et al. 2013). We will certainly include this possibly not ideal choice in the discussion section of our result manuscript. 

Comment 6: Her is another confusing problem for the prognostic tool you are about to produce and please allow me to elaborate and explain my worries and its all centered around Kinesiophobia. Kinesiophobia defined as “an excessive, irrational, and debilitating fear of physical movement and activity resulting from a feeling of vulnerability due to painful injury or reinjury”. It is by itself a central factor in the process of pain developing from acute to chronic stages, The Cognitive Fear Avoidance Model describes that very clearly, it also linked it strongly to catastrophizing, re/injury, avoidance behaviour, and on the long run inactivity, disability, depression, low self-esteem and increasing the risk for a wide range of health problems, functional decline and premature death.

My question is: You have included so many independent factors that have high potential prognostic ability alone, so it become very difficult and confusing to understand the weighting of theses variables to the final prediction scores on your prognostic tool while in reality each one of which is so representative and highly weighted predictor itself such as kinesiophobia , depression, anxiety, low self-esteem. Need to elaborate a bit more on this.

If we understand this question correctly, a part of this question is answered under your comment 1.’

In addition, we also include the potential multi-collinearity between different psychological constructs in our statistical analysis. We have now also described this in our statistical analysis section (page 18, lines 316-318):

Multicollinearity between candidate predictors will be assessed using the variance inflation factor. In case the variance inflation factor exceeds 10, we will select which candidate predictor add to the modeling phase based on clinical expertise. 

Thank you for this relevant point about the concrete weighting of the different psychological factors in our prognostic tool. We will certainly include this comment and critically reflect on our outcomes in our future result paper.

Comment 7: Discussion

Line 338-350 I found the following statement very confusing and not helpful to justify the objectives of the study:

- 338. The given therapy may influence the outcome and the accuracy and transportability of the model to be developed.

- 340. The patients receive standard care based on the Dutch Physiotherapy Guideline for neck pain.

- 344-345. We will have no information on this form of treatment during this study; however, it could influence the outcome.

You wrote “Nevertheless, we consider the impact on our study findings to be minimal, given ……..”

In their conclusion Carroll LJ, Hogg-Johnson S, van der Velde G, Haldeman S, Holm LW, Carragee EJ, Hurwitz EL, Côté P, Nordin M, Peloso PM, Guzman J, Cassidy JD. Course and Prognostic Factors for Neck Pain in the General Population: Results of the Bone and Joint Decade 2000 –2010 Task Force on Neck Pain and Its Associated Disorders. Eur Spine J. 2008 Apr;17(Suppl 1):75–82. doi: 10.1007/s00586-008-0627-8. Epub 2008 Feb 29. PMCID: PMC2271093. “General exercise was not prognostic of better outcome; however, several psychosocial factors were prognostic of outcome”.

In their discussion they also have specifically stated that “ Prognosis may also depend on whether or not the exercises themselves were designed to impact the neck and shoulder areas”.

My question since the author of the above study concluded only on general exercises, but have raised the potential value of specific designed exercise then this raise the concerns about the type of approach used in these multicenter and by the physiotherapist skills and preferences involved in this study. 

The benefit and impact of specific approaches as the directed self treatment exercise for the neck used by well trained therapist such as Maitland, McKenzie, or Shacklock M neurodynamic and others on the outcomes has ben well documented such techniques has prognostic value against psychological factors and chronicity.

For example directional preference exercise by the McKenzie approach help the patient face his one fear of painful movement direction while rehearsing in this direction which help centralized improve and prevent the next episodes of pain and fast return to work.

Line 348-350

You wrote

“Thereby, we will report the physiotherapy treatment the patient received and discuss the possible impact on our study findings (TRIPOD 5C) but do not include the different treatments as a predictor in our model” .

I think you may need to reconsider these statement in light of the variation of approaches among theses centers to reveal the impact of these specific neck exercise approach on modifying the psychological factors

Although we agree with your statement here, this would require a different study design to assess the influence of various PT approaches on psychological variables.

As described earlier in comment 5, achieving a reliable sample size in a controlled manner is challenging in such a specific research group. We had to make choices based on that. Unfortunately, we were unable to includer the various treatments applied. In addition, if our study is very homogeneous, we will get that back in our external validation. Therefore, we have described in our discussion that we are aware that this may modify the modifiable factors included. However, we give reasons why this effect on our results may also be limited in our discussion section (page 21, lines 368-376):

We will have no information on this form of treatment during this study; however, it could influence the outcome. Nevertheless, we consider the impact on our study findings to be minimal, given (1) the heterogeneity of the factors to be modified, (2) the multiple modalities used by physiotherapists, and (3) the difference in physiotherapists’ backgrounds. Thereby, we will report the physiotherapy treatment the patient received and discuss the possible impact on our study findings (TRIPOD 5C) but do not include the different treatments as a predictor in our model. Moreover, the current setting does reflect clinical practice as it is. This heterogeneity is likely to remain even after implementing of a well-performing model. 

We included the physical therapist's variable attitude as a candidate prognostic factor. This will partly obviate that the biopsychosocial therapist may pay more attention to psychosocial factors, and the 'biomedical' therapist pays more attention, for example, to the manual therapeutic techniques you have mentioned.

Reviewer 4

I would like to thank the authors for their time and efforts in drafting this protocol study entitled, “A study protocol for the validation of a prognostic model with an emphasis on modifiable factors to predict chronic pain after a new episode of acute- or subacute nonspecific idiopathic, non-traumatic neck pain presenting in primary care”. I found it to be of interest and very well written. 

Comment 1: However, the authors may consider clarifying if the participants in their study will be asked to report pain intensity “in the last 24 hours” or “an average pain intensity” because recall of pain may vary.

Thank you for this comment. See our reaction to comment 5 of reviewer 2.

Comment 2: I strongly suggest that the authors include the operational definitions for acute and subacute neck pain knowing that determining whether a patient fits into the acute or subacute time-period is not always as precise as we would like to think. 

We are aware of the wide variance in categorizing acute and sub-acute neck pain. In the literature, we found categorization between <1 week to < 6 weeks for acute neck pain. Most studies comparing the outcomes between the subgroups (acute vs. subacute) handle this variable as a dichotomous variable. In our opinion, there is no hard cut-off point in the duration of the pain and the different outcomes of treatment or prognosis; that is why we handle the duration of neck pain as a continuous variable in our statistical analysis. 

We also add that in our manuscript under the description of the measurement ‘symptoms’:

(page 13, lines 181-182) Duration of pain will be handled as a continuous variable in our statistical analysis since there is no hard cut-off point between ‘acute’ and ‘sub-acute’ pain.

However, we use the terms' acute' and 'sub-acute' in our study descriptions, so a more precise description of our categorization is recommended. In our introduction and participants' descriptions, we clarify this is as follows: 

(page 5, lines 87-90) Therefore, there is a need for a prognostic study that identifies modifiable prognostic factors using a biopsychosocial view, that includes only patients with acute- (0 to 3 weeks) or subacute (4 to 12 weeks) nonspecific idiopathic, non-traumatic neck pain, to help prevent chronification of pain in physiotherapy practices.

(page 7, lines 119-120) The patients will be approached if they present with a new episode of acute- (0 to 3 weeks) or subacute (4 to 12 weeks) nonspecific idiopathic, non-traumatic neck pain. 

Comment 3: In addition, I recommend including the participants’ number of previous episodes of neck pain as a factor in the analysis because number of previous episodes is associated with subsequent outcome in musculoskeletal pain symptom. 

Thank you for this suggestion. In our questionnaire, we asked the patients when they experienced the last episode of neck pain. However, patients indicate that they do not know precisely when the last episode was. Besides, it is known that it is challenging to recall both the pain variation and intensity dimensions of neck pain (Irgens et al., 2022). Indicating the number of previous episodes of neck pain by the patients will be even more difficult for patients with more than one episode and is thereby also not a reliable factor for our analysis. 

Comment 4: Although the time-period “0–4 weeks of symptoms” is well-known definition of “acute” neck pain, during analysis, I suggest (if applicable) exploring the relationship between participants prognostic factors at baseline and outcome for a subgroup of patients with acute neck pain with shorter period (0–2 weeks) “high-acute” and (2-4) “mid-acute” of symptoms.

Thank you for this comment. See our reaction on ‘comment 2’.

Comment 5: Lastly, one formatting note: I believe the full stop should be placed after the in-text citation.

Thank you for making us aware of this. Since we are using the Vancouver referencing style, the full stop should be placed after the in-text citation. We changed this in our manuscript.

---

## [Decision Letter · Decision Letter 1]

26 Dec 2022

A study protocol for the validation of a prognostic model with an emphasis on modifiable factors to predict chronic pain after a new episode of acute- or subacute nonspecific idiopathic, non-traumatic neck pain presenting in primary care.

PONE-D-22-19562R1

Dear Dr. Verwoerd,

We’re pleased to inform you that your manuscript has been judged scientifically suitable for publication and will be formally accepted for publication once it meets all outstanding technical requirements.

Kind regards,

Aqeel M Alenazi

Academic Editor

PLOS ONE

Additional Editor Comments (optional):

Thank you for addressing the comments and suggestions. Now the paper is accepted.

Reviewers' comments:

Reviewer's Responses to Questions

**Comments to the Author**

1. Does the manuscript provide a valid rationale for the proposed study, with clearly identified and justified research questions?

Reviewer #2: Yes

Reviewer #3: Yes

Reviewer #4: Yes

2. Is the protocol technically sound and planned in a manner that will lead to a meaningful outcome and allow testing the stated hypotheses?

Reviewer #2: Yes

Reviewer #3: Yes

Reviewer #4: Yes

3. Is the methodology feasible and described in sufficient detail to allow the work to be replicable?

Reviewer #2: Yes

Reviewer #3: Yes

Reviewer #4: Yes

4. Have the authors described where all data underlying the findings will be made available when the study is complete?

Reviewer #2: No

Reviewer #3: Yes

Reviewer #4: No

5. Is the manuscript presented in an intelligible fashion and written in standard English?

Reviewer #2: Yes

Reviewer #3: Yes

Reviewer #4: Yes

6. Review Comments to the Author

You may also provide optional suggestions and comments to authors that they might find helpful in planning their study.

Reviewer #2: Thanks for addressing my comments. I hope you found the reviews helpful and wish you good luck with conducting the study.

Reviewer #3: I thank you the authors for their responses to my criticism and I found that all their responses to be rational and valid

Reviewer #4: A comprehensive study and will provide valuable data for the prognosis of chronic pain among patients with acute and subacute neck pain.

7. PLOS authors have the option to publish the peer review history of their article (what does this mean?). If published, this will include your full peer review and any attached files.

Reviewer #2: **Yes: **Alice Kongsted

Reviewer #3: **Yes: **professor Saud M. Al-Obaidi Ph.D PT

Reviewer #4: No

---

## [Editor Report · Acceptance letter]

3 Jan 2023

PONE-D-22-19562R1 

A study protocol for the validation of a prognostic model with an emphasis on modifiable factors to predict chronic pain after a new episode of acute- or subacute nonspecific idiopathic, non-traumatic neck pain presenting in primary care. 

Dear Dr. Verwoerd:

I'm pleased to inform you that your manuscript has been deemed suitable for publication in PLOS ONE. Congratulations! Your manuscript is now with our production department. 

Kind regards, 

on behalf of

Dr. Aqeel M Alenazi 

Academic Editor

PLOS ONE